# Ultraprecise Detection of Influenza Virus by Antibody-Modified Graphene Transistors

**DOI:** 10.3390/s25030959

**Published:** 2025-02-05

**Authors:** Gang Wang, Mingming Zhang, Minghua Zhu, Tengfei Zhang, Xueqin Qian, Yili Liu, Xinye Ma, Changhao Dai, Dacheng Wei, Zhaoqin Zhu, Juntao Sun, Mingquan Guo

**Affiliations:** 1Department of Laboratory Medicine, Shanghai Public Health Clinical Center, Fudan University, Shanghai 201508, China; 2State Key Laboratory of Molecular Engineering of Polymers, Department of Macromolecular Science, Fudan University, Shanghai 200438, China; 3Department of Gastroenterology, The Second Hospital, Cheeloo College of Medicine, Shandong University, Jinan 250012, China

**Keywords:** influenza, antigen test, accuracy, field-effect transistor

## Abstract

Over the past decade, the large-scale spread of influenza viruses has posed an increasing burden on public health. The effective screening of influenza agents requires a fast, precise, on-site and easy-to-operate method. Unfortunately, current screening methods face challenges in speed and accuracy, especially in complex on-site settings. Here, this work develops a nucleoprotein antibody-modified graphene field-effect transistor (NPAb-GFET) for rapid and highly precise detection of influenza A viruses. The functionalized monoclonal antibodies capture influenza virus nucleoprotein within 100 × 10^−9^ s on the sensing surface. Therefore, the developed NPAb-GFET achieves an average response time of 72.1 s when detecting influenza A viruses in clinical samples. Furthermore, the testing of 106 throat swab samples exhibits an accuracy of 99.1%. This finding provides a valuable diagnostic tool for the control of influenza viruses, accelerating the population-wide control of other epidemics.

## 1. Introduction

The World Health Organization (WHO) statement on clusters of respiratory diseases among children notes a sudden increase in outpatient and inpatient visits to children due to infectious agents, including influenza virus, respiratory syncytial virus, adenovirus and Mycoplasma pneumoniae [1,2,3]. Notably, influenza is an acute respiratory infectious disease caused by influenza A or B viruses and occasionally by influenza C viruses [4,5]. The incubation period of influenza virus infection is 1~4 days (average 2 days), usually accompanied by high fever, sore throat, chills, headache, vomiting, secondary bacterial infection, pneumonia, encephalitis, etc. [6,7]. Considering the fact that the influenza A virus has many hosts in nature and is more likely to mutate or reassort, resulting in its rapid spread in the human population, multiple large-scale outbreaks of influenza have historically been associated with influenza A [8]. Therefore, rapid and accurate early detection methods are of vital importance for influenza disease control.

Clinically, existing laboratory influenza tests include antigen testing, polymerase chain reaction (PCR)-based nucleic acid testing and serum antibody testing [9,10]. A commonly used method for influenza antigen–antibody detection is the colloidal gold assay technique; although it is simple and low-cost, the disadvantages are the relatively low sensitivity, high leakage rate and susceptibility to false-negative results. This may be related to the unstable viral load of nasal swab specimens and the low sensitivity of the colloidal gold methodology [11,12]. Nucleic acid testing is the use of reverse transcription quantitative polymerase chain reaction (RT-qPCR) to detect influenza A/B virus; this method has higher sensitivity than antigen detection, allows for relatively quantitative detection and can distinguish typing. The disadvantages are that the operation takes longer (about 5–6 h), the results are greatly affected by the sampling technique and the laboratory equipment and application scenario requirements are higher [13,14]. Therefore, there is an urgent need for a portable, high-speed, high-sensitivity laboratory antigen detection technique for influenza testing.

Electrochemical biosensors have attracted enormous attention because of their simple operation, low cost and high response speed in complicated on-site settings [15]. To date, notable advances have been made in developing various electrochemical biosensors for infectious diagnoses, including electrochemiluminescence biosensors, transistor-based biosensors, amperometric biosensors and so on [16,17]. Among them, graphene field effect transistor sensors (GFETs), sensors using graphene as an active semiconductor material, are rapidly developing for biosensing applications. GFETs mainly rely on the principle that when biomolecules interact with the graphene surface, the electronic state of graphene is altered. This change leads to a shift in the Fermi level and, consequently, a change in the position of the Dirac point. By detecting this Dirac point shift, the detection and analysis of biomolecules can be achieved [18,19,20]. The characteristic curves of GFETs have a good linear relationship, which reflects the real-time current signal response of the device to electrochemical and electrophysical reactions occurring on the graphene surface, and the response increases with the analyte concentration [21,22]. In 2014, Li et al. developed a multifunctional ultrasensitive immunosensor for the detection of H1N1 hemagglutinin (HA), which can detect antigen concentrations as low as 10^−13^ g mL^−1^ [23]. Afterward, Wu et al. achieved a limit of detection (LoD) as low as 125 copies mL^−1^ using an MXene–graphene field effect transistor (FET) sensor [24]. The other study was on an anti-hemagglutinin (anti-H1)-based electrochemical immunosensor for H1N1 detection in synthetic saliva, with a LOD as low as 15 µg mL^−1^, but the detection technique requires 1 h of incubation pretreatment time [25]. However, existing electrochemical biosensors face a trade-off between testing accuracy and speed of influenza screening, especially in complicated on-site settings.

Herein, we demonstrate a nucleoprotein antibody-modified graphene field-effect transistor (NPAb-GFET, Figure 1a–c) for the rapid and ultrasensitive detection of influenza A viruses. The nucleoprotein antibodies functionalized on the graphene surface were able to capture influenza A viruses within 100 × 10^−9^ s, leading to an average response time of 72.1 s. Furthermore, the developed NPAb-GFET has an LoD down to 30 copies mL^−1^ and an accuracy of 99.1% in 106 clinical tests. Owing to its high accuracy and response speed, the NPAb-GFET is a valuable tool for precise diagnosis and epidemic control in the future.

## 2. Materials and Methods

### 2.1. Materials

Bovine serum albumin and polymethyl methacrylate were purchased from Sigma-Aldrich, St. Louis, MO, USA. Polydimethylsiloxane (PDMS) was from Shenzhen Oswang Co., Ltd., Shenzhen, China. Influenza A virus nucleoprotein antibody (GTX629633, 1:1000) was purchased from GeneTex, Irvine, CA, USA. S1813 and LOR 3A photoresists were from Xi’an Bynano Co., Ltd., Xi’an, China. Copper foil (25 μm) was from Thermo Fisher Scientific Inc., Waltham, MA, USA. Graphene was prepared by reported methods [26,27]. The 1× phosphate-buffered saline (PBS, pH 7.4) was purchased from Sino Biological, Inc., Beijing, China. Ultrapure water (18.2 mΩ cm) was obtained from Millipore equipment (Milli-Q™ Direct Water Purification System, Merck KGaA Darmstadt, Germany). All the reagents and solvents were of analytical purity and were used without further purification.

### 2.2. Device Fabrication and Characterization

The copper foil-grown graphene was transferred to a SiO_2_/Si (300 nm oxide) substrate of approximately 2 cm × 2 cm by an electrochemical bubbling method. Then, GFETs with channel sizes of 30 μm × 300 μm were fabricated via standard photolithography on the SiO_2_/Si sheet [28]. PASE was used as a linker molecule modified on the graphene surface by non-covalent coupling, after which the GFET device was immersed in the synthesized specific antibody solution and incubated overnight. To validate the immobilization of antibodies on the surface, X-ray photoelectron spectroscopy (XPS) and Transfer Curve Characterization were utilized to detect the modification of the probe at the graphene interface [29]. Raman spectroscopies and mapping analysis were conducted with a Raman spectrograph (XploRAH, ORIBA Jobin Yvon, Glasgow, UK) with a 532 nm laser (20 mW). Additionally, the morphology of the antibody-modified graphene sensing interface was characterized by atomic force microscopy (AFM, Cypher VRS1250, Oxford Instruments, Abingdon, UK) in liquid-sweep mode [30].

### 2.3. Electric Tests

The developed NPAb-GFETs were connected to a Keithley 2612B semiconductor analyzer (Figure 1c). The PDMS well was filled with PBS buffer to cover the graphene channel, the corresponding electrical signals gradually changed and stabilized, and the electrical signals returned to the baseline position. When the baseline is stable, the influenza antigen protein mixture at the corresponding concentration is infused into the flow channel at a set flow rate. The parameters tested were Vds = 100 mV and Vgs = −150 mV to 450 mV. For the Ids–Vds tests, Vds was set from −100 mV to 100 mV, and Vgs was 0 mV. After the construction of the GFET biosensor, we evaluated the graphene uniformity of the device in the solution system by testing the transconductance of graphene. Since there exists an ion exchange process between the Ag/AgCl reference electrode and the solution system, the gate voltage applied via the Ag/AgCl reference electrode experiences no voltage drop [28]. One-way analysis of variance (ANOVA) and multiple comparison tests were performed to compare differences between groups. The statistical details can be found in the figure legends.

### 2.4. Analyte Binding Analysis

For protein docking, the X-ray crystal structures of 6I54 and the antibody (2RGS) were retrieved from the Protein Data Bank. The Docking Web Server (GRAMM) was applied for protein–protein docking, which was optimized by AutoDockTools-1.5.7. During the binding process of an antibody to an antigen, the structure of the antibody undergoes dynamic changes. Utilizing Root Mean Square Deviation (RMSD) and Root Mean Square Fluctuation (RMSF) analysis can reveal the structural stability and atomic flexibility changes in the variable region (antigen-binding site) of the antibody before and after binding. By calculating RMSD, one can determine the degree of change in the overall structure of the antibody during the binding process. Meanwhile, RMSF analysis can assist in identifying which amino acid residues in the variable region have flexibility crucial for binding affinity [31]. The molecular dynamics simulations were running for 100 ns at 300 K.

### 2.5. Clinical Sample Testing

To evaluate the practical feasibility of the developed NPAb-GFET, 106 clinically isolated throat swab samples were collected for clinical validation (see Appendix A Appendix A). The clinically applied testing process protocols are shown in Appendix A Appendix A. The graphene sensing interface was immersed in 1× PBS buffer to maintain the bioactivity of the sensing interface. The corresponding subject throat swab sample buffer (20 μL) was added to the channel of NPAb-GFETs. When it flows to the graphene channel, the viral protein will specifically bind to the antibody probe at the graphene interface and produce the current response signal. When the binding and dissociation of the protein to the antibody reach equilibrium, the inverse electrical response signal changes are generated. The flow rate in the microchannel was constant throughout the test, and the set parameters were VDS = 50 mV and VGS = 0 V. As described in our previous research, in principle, for each device, we performed alternating detections of positive and negative samples with clinically known test results to avoid the interference effects of strongly positive samples [28]. All human participants in this work were reviewed and approved by the Ethics Committee of Shanghai Public Health Clinical Center Affiliated with Fudan University (#2022-Y025-01). Written informed consent was obtained from the individual(s) for the publication of any potentially identifiable images or data included in this article.

## 3. Results and Discussion

### 3.1. Functionalization of NPAb-GFET

To validate the antibody functionalization of the GFETs, we tested their transfer curves in 1× PBS buffer. The Dirac point (*V*_Dirac_) of the NPAb-GFETs positively shifted 8~20 mV after the immersion of PASE and influenza A virus nucleoprotein antibody (Figure 2a). This is attributed to the π–π interaction between PASE and graphene, which proves the correctness of the functionalization. After the immobilization of mAbs, the N 1s peak obtained by X-ray photoelectron spectroscopy shifted in both intensity and position (Figure 2b). The successful antibody functionalization can also be characterized by AFM (Figure 2c) and Raman results (Figure 2d and Appendix A). The uniformity of the graphene in the device was evaluated by measuring the transconductance capacitance of graphene. We found that when the applied gate voltage Vgs was set at 0 mV, the absolute value of the graphene transconductance gm was approximately 200 µS (Appendix A). This indicates that the graphene synthesized by the electrochemical bubbling method in this study has very high mobility and can meet the detection requirements for low-concentration targets in complex biological fluids.

Furthermore, we analyzed the docking process of antigen–antibody molecules to clarify the characteristics and interactions of antigen–antibody complexes. In the simulations, the antigen–antibody complex was formed mainly on three sets of hydrogen bonds to maintain a more stable structure (Figure 3a). The RMSD of the antigen–antibody complex gradually stabilized at 2.0~4.0 Å after 100 ns (Figure 3b), and the RMSD of the protein amino acid residues in the antigen–antibody complex gradually converged from 4 to 8 Å (Figure 3c). The RMSF of protein amino acid residues in the antigen–antibody complex was also verified by the gradual convergence of the RMSF from 4 to 6 Å (Figure 3c). The RMSD demonstrated that the antigen–antibody can form a structurally stable complex. Meanwhile, the results of RMSF proved that the amino acids of the antibody molecule have high flexibility, which contributes to the intimate binding of the antibody probe to the target antigen. These results indicate that nucleoprotein antibodies immobilized on graphene can capture influenza antigens effectively [32,33].

### 3.2. Detection Mechanism

The transfer curves (*I*_ds_–*V*_gs_) of the NPAb-GFET were tested on simulated samples of antigen proteins in 1× PBS. In samples with concentrations ranging from 0 to 10^5^ copies mL^−1^, the NPAb-GFET exhibited substantial negative Dirac shifts (Appendix A). This can be ascribed to the n-doping effect when the nucleoprotein antibody and target antigens bind to the graphene surface [26]. Owing to its atomic thickness, the NPAb-GFET is extremely sensitive to external perturbations and can detect as low as 30 copies mL^−1^ influenza A proteins. When the binding and dissociation of the protein to the antibody reach equilibrium, the inverse electrical response signal changes are generated.

In addition to transfer curves, the *I*_ds_-t curves of the NPAb-GFETs are used to monitor the sensing processes in real time. The Δ*I_ds_*/*I*_ds0_ response of the sensor tends to increase as the concentration of the target H1N1 antigen solution increases. The straight line and error bars fitted to the Δ*I_ds_*/*I*_ds0_ response values of the viral antigen analytes at different concentrations also indicate a good linear relationship after fitting the test data (Appendix A).

### 3.3. Clinical Tests

To verify the sensing performance in clinical samples, we tested 106 clinical samples with the NPAb-GFETs (Figure 4a). The initial current fluctuation (i.e., Δ*I*_ds_/*I*_ds0_) of the NPAb-GFETs is recorded as the baseline when no sample is added. Three times the baseline value is considered as the detection threshold for determining the presence of the target pathogen in the sample to be tested, and the time taken for Δ*I*_ds_/*I*_ds0_ to reach the detection threshold is the response time of the sensor [34]. The results revealed that influenza-negative clinical samples N6 and N7 elicited a Δ*I*_ds_/*I*_ds0_ of less than 0.05%, whereas positive clinical sample P5 elicited a Δ*I*_ds_/*I*_ds0_ of more than 6%, and the detection time was shorter than 100 s (Figure 4b and Appendix A). After all 32 positive clinical samples were tested, the average response time was 72.1 s (Figure 4c), which is 10–800 times faster than various other immunofluorescence or PCR assays (Appendix A) [25,35,36,37,38,39,40,41,42,43,44]. This negative shift is also verified by the negative shift in ΔIds in power versus sample size curves (Appendix A). The statistical power increases little when the sample size is larger than 100, which also proves the suitability of the device.

For statistical analysis, the real-time current response values of all the clinical samples revealed that the NPAb-GFET can sensitively detect antigens in positive samples (P1~P32) with relatively large Δ*I*_ds_/*I*_ds0_ values (>1.0%). In comparison, the NPAb-GFET has a small Δ*I*_ds_/*I*_ds0_ (<0.2%) in negative samples (N1~N38) and healthy samples (H1~H36). This further demonstrates that the NPAb-GFET can quickly and clearly distinguish between positive and negative samples. Furthermore, we obtained receiver operating characteristic (ROC) curves (Figure 4e) and a confusion matrix (Figure 4f) via computational fitting of the test data. The results indicate that NPAb-GFETs have an overall accuracy of 99.1%, with 98.6% sensitivity and 97.0% specificity in 106 clinical tests. Through extensive literature research, we have obtained various emerging clinical pathogen molecular detection technologies in recent years, including CRISPR-Cas13a, FICT, MNPs, ELISA, SERS-LFA, qPCR, LAMP, thermophoretic assays and so on (Appendix A). The high accuracy is not a trade-off of testing speed, making our NPAb-GFET a promising tool in influenza tests (Figure 4g) [23,25,35,44,45,46,47,48,49].

## 4. Conclusions

In conclusion, we developed an NPAb-GFET for the accurate detection of influenza A virus in complicated on-site environments. The developed NPAb-GFET can detect down to 30 copies mL^−1^ influenza A proteins in 1× PBS. Owing to its high sensitivity, the NPAb-GFET is highly accurate (99.1% overall accuracy) and has a short testing time (72.1) in 106 clinical tests. Therefore, this work provides an accurate, rapid, low-cost and easy-to-operate method for the detection of influenza viruses, which has promising potential in infectious screening and epidemic control.

## Figures and Tables

**Figure 1 sensors-25-00959-f001:**
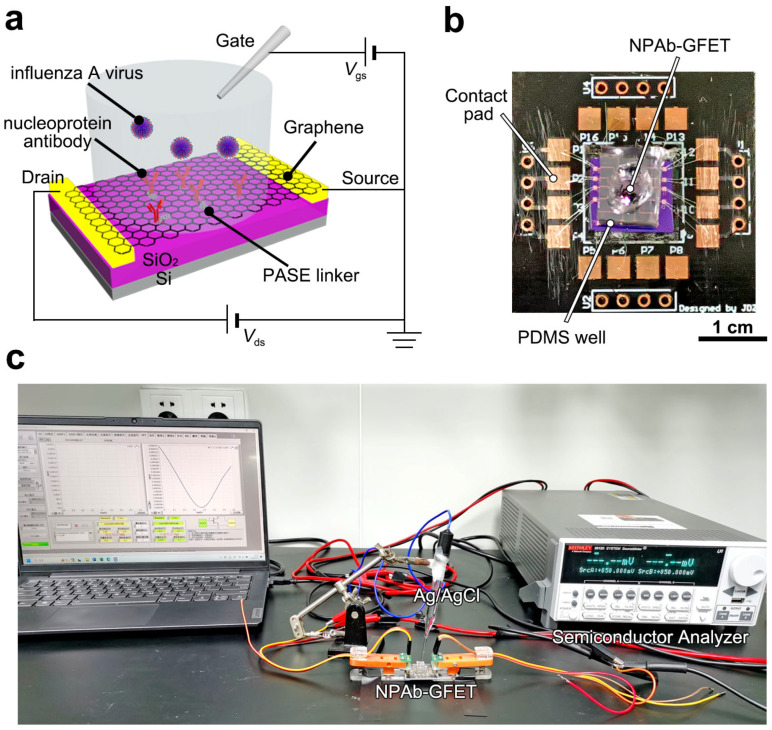
Device configuration of NPAb-GFET sensors. (**a**) Schematic diagram of the NPAb-GFET. (**b**) Image of a packaged NPAb-GFET sensor on a printed circuit board, where the PDMS well is set to hold analyte solution. (**c**) Image of the electrical measurement setup.

**Figure 2 sensors-25-00959-f002:**
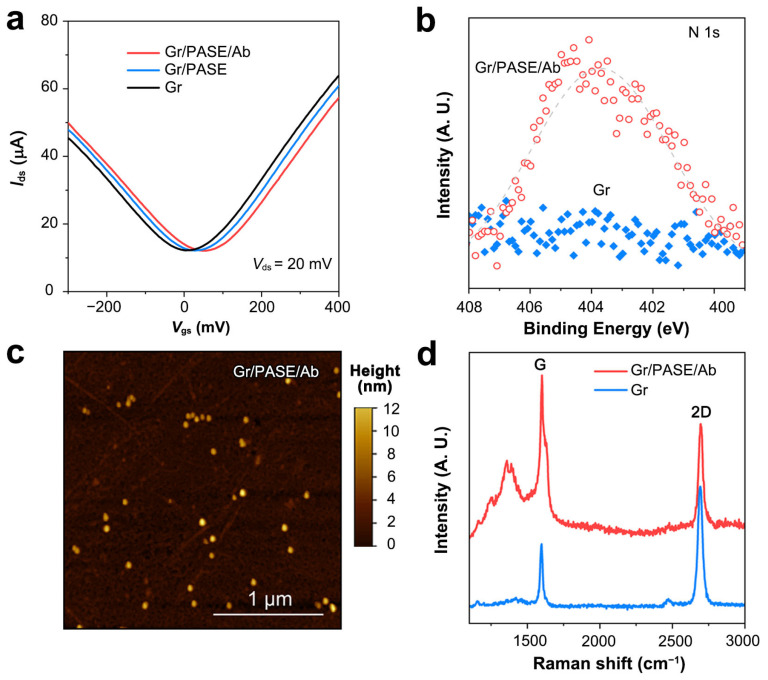
Characterization of NPAb-GFET sensors. (**a**) Transfer curves of the pristine, PASE-modified and antibody-modified GFETs. (**b**) N 1s region for pristine and antibody-modified graphene. (**c**) AFM image of the antibody-modified graphene surface. (**d**) Raman spectra of pristine graphene and antibody-modified graphene.

**Figure 3 sensors-25-00959-f003:**
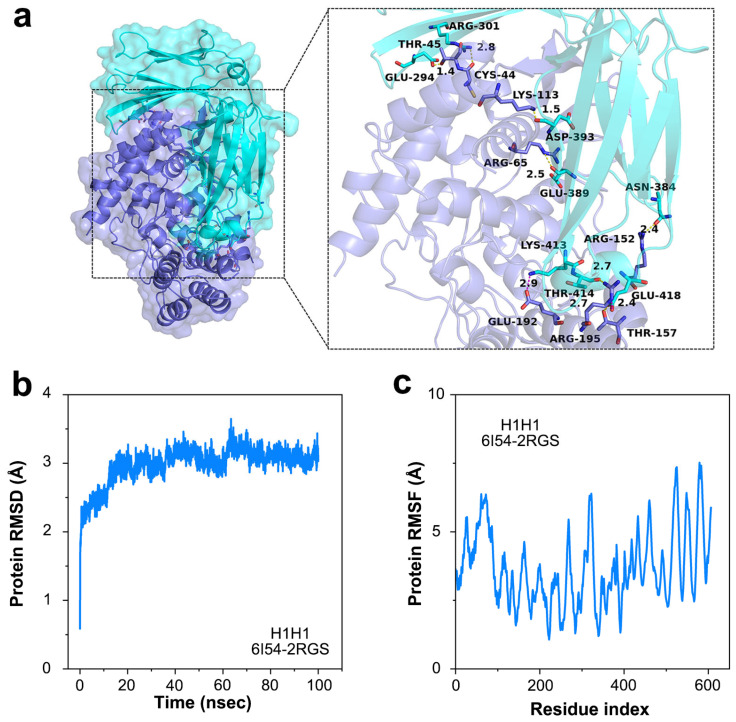
Binding process on the sensing surface of NPAb-GFET. (**a**) The protein–protein complex (6I54-2RGS) demonstrating the binding sites of influenza A virus. (**b**) The protein RMSD versus time curve of the protein–protein docking. (**c**) The protein RMSF versus time curve of the protein–protein docking.

**Figure 4 sensors-25-00959-f004:**
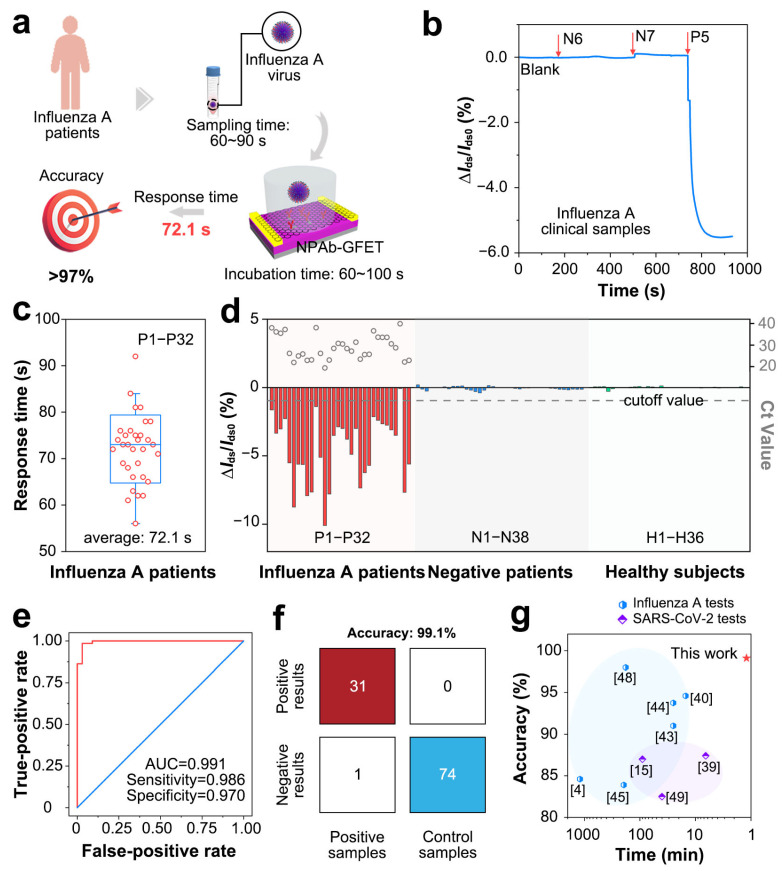
Clinical performance of NPAb-GFET sensors. (**a**) Schematic diagram for detecting influenza A virus via NPAb-GFET sensor. (**b**) ΔIds/Ids0 versus t curve upon addition of influenza negative samples (N6 and N7) and positive sample P5, response values of |∆Ids/Ids0|(%) = |(Ids−Ids0)/Ids0| × 100%, where Ids is the real-time drain-source current and Ids0 is the initial current, Vd is source-drain voltage, Vg is gate voltage. (**c**) Box plot of the diagnoses time for P1–P32. (**d**) ΔIds/Ids0 upon addition of 106 clinical samples containing P1–P32, N1–N38 and H1–H36. The gray dashed line indicates the cutoff value applied in the diagnoses. (**e**) Receiver operating characteristic (ROC) curve for influenza A diagnosis. AUC represents area under the curve. (**f**) A confusion matrix of the NPAb-GFET. (**g**) Comparison of current laboratory tests and commercial kits for accuracy.

## Data Availability

Data is contained within the article or Appendix A.

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
