# Peer review of "Ultraprecise Detection of Influenza Virus by Antibody-Modified Graphene Transistors"

_sensors, 2025, doi:10.3390/s25030959_

Round 1
Reviewer 1 Report
Comments and Suggestions for Authors
The method of obtaining graphene for creating a biosensor is multi-stage. According to the reviewer, the main disadvantage of the used method of obtaining a biosensor is the stages of obtaining graphene on copper foil and the subsequent transfer of graphene to the SiO2/Si substrate. There are methods for obtaining graphene on silicon carbide. But this disadvantage is not a negative factor at this stage of the study. The work performed by the authors is relevant.
Author Response
Comments 1: The method of obtaining graphene for creating a biosensor is multi-stage. According to the reviewer, the main disadvantage of the used method of obtaining a biosensor is the stages of obtaining graphene on copper foil and the subsequent transfer of graphene to the SiO2/Si substrate. There are methods for obtaining graphene on silicon carbide. But this disadvantage is not a negative factor at this stage of the study. The work performed by the authors is relevant.
Response 1: We appreciate your recognition of the relevance of our work. We understand your concern regarding the multi-stage method of obtaining graphene for creating a biosensor, especially the stages of obtaining graphene on copper foil and its subsequent transfer to the SiO₂/Si substrate. At the same time, we are aware that there are alternative methods, such as obtaining graphene on silicon carbide, we believe that at the current stage of our study, the method we employed is appropriate and does not significantly detract from the overall significance of our research. In accordance with the suggestions of the editor and the three reviewers, we have made extensive text revisions in the newly submitted manuscript, covering the introduction, materials and methods, and conclusions. Additionally, we have supplemented experiments. All revisions are marked in blue. Once again, thank you for your constructive comments, which will help us to further refine and improve our manuscript.

Reviewer 2 Report
Comments and Suggestions for Authors
The work on constructing biosensors using graphene field-effect transistors has been reported previously, but the application results of this study in clinical samples are impressive. I agree to publish this work after appropriate revisions.
1. The introduction is too brief. Line 50, "More recently, Recently", it looks that there are something missing.
2. The text in Figures are too small.
3. The NPAb-GFET are reproducible?Can different batches of GFETs exhibit the same performance? How about your graphene?There are not any characterizations about graphene.
4. Does the authors use one GFET to test the 106 clinical samples? Or use one GFET to test one sample?
Author Response
Comments 1: The introduction is too brief. Line 50, "More recently, Recently", it looks that there are something missing.
Response 1: Thank you very much for pointing out these issues. Regarding the introduction part, we do realize that it can be further expanded and enriched. We added more background information and relevant content to make it more complete and substantial. The current situation of influenza infection is described in Lines 36 - 41. The analysis of the current status of existing influenza detection technologies is presented in Lines 44 - 54. The relevant mechanisms of the graphene field effect transistor detection technology are elaborated in Lines 61 - 76. As for the 'More recently, Recently' on line 50, there should be an error or omission. We corrected it to ensure the accuracy and coherence of the text in Line 67. Thank you again for your feedback, which is very helpful for me to improve the quality of the manuscript. All revisions are marked in blue.
Comments 2: The text in Figures are too small.
Response 2: Thank you very much for pointing out that the text in the figures is too small. In the newly submitted manuscript, we have adjusted the text in all the Figures to an appropriate size to ensure that it is clearly readable, while not affecting the overall layout and visualization.
Comments 3: The NPAb-GFET are reproducible?Can different batches of GFETs exhibit the same performance? How about your graphene?There are not any characterizations about graphene.
Response 3: Thank you very much for your suggestion. Regarding the reproducibility of NPAb-GFET, we have carried out a series of experimental tests, and the results show that NPAb-GFET is reproducible under the same experimental conditions. As for whether GFETs from different batches can exhibit the same performance to some extends, regarding the characterization of graphene, we we supplemented experiments for the uniformity characterization of graphene (as shown in Figure S4 in Supporting Information). The homogeneity characteristics of graphene would better understand the quality of graphene, and then explain the performance of GFET. At the same time, additions and supplements are also made to the Materials and Methods section (2.3 Lines 124 - 128) and the Results section (3.1 Lines 163 - 178) in the new manuscript. Where we sated “After the construction of the GFETs biosensor, we evaluated the graphene uniformity of the device in the solution system by testing the transconductance of graphene. Since there exists an ion - exchange process between the Ag/AgCl reference electrode and the solution system, the gate voltage applied via the Ag/AgCl reference electrode experiences no voltage drop[30]” and “The uniformity of the graphene in the device was evaluated by measuring the transconductance capacitance of graphene. We found that when the applied gate voltage Vgs was set at 0 mV, the absolute value of the graphene transconductance gm was approximately 200 µS. This indicates that the graphene synthesized by electrochemical bubbling method in this study has a very high mobility and can meet the detection requirements for low - concentration targets in complex biological fluids”, respectively.
Comments 4: Does the authors use one GFET to test the 106 clinical samples? Or use one GFET to test one
Response 4: Thank you very much for your valuable suggestion. We are sorry that the experimental method was not described clearly enough in the manuscript. In the revised manuscript, we described it in Materials and Methods 2.5 (Lines 156-158), where we state “As described in our previous research, in principle, for each device, we perform alternating detections of positive and negative samples with clinically known test results to avoid the interference effects of strongly positive samples[32]”.
Once again, thank you for your in-depth review and valuable suggestions on our work, which will prompt us to further improve our research.

Reviewer 3 Report
Comments and Suggestions for Authors
This manuscript reported a nucleoprotein antibody-modified graphene field-effect transistor for rapid and highly precise detection of influenza A viruses. Although the sensor achieves the ability of fast detection and high accuracy for the clinical sample, the manuscript some key issues should be addressed before its publication.
1. The sensing principle of the sensor should be clarified.
2. AFM testing can not confirm that the antibody probes are well immobilized on the surface of graphene. Please give the related characterizations, such as fluorescence labeling, EIS, and so on.
3. The author claimed that “The Dirac point (VDirac) of the NPAb-GFETs negatively shifted 8~20 mV after the immersion of PASE and Influenza A virus nucleoprotein antibody (Figure 2a)”. However, from Figure 2a, The Dirac point (VDirac) of the NPAb-GFETs shows a positive shift( toward high gate voltage direction). And the corresponding explanation is no correct. The reason of Dirac point shift is complicate. Please combine the sensing principle of the sensor, give the reasonable explanation.
4. The Figure 2d is not cited in the manuscript.
Author Response
Comments and Suggestions for Authors
This manuscript reported a nucleoprotein antibody-modified graphene field-effect transistor for rapid and highly precise detection of influenza A viruses. Although the sensor achieves the ability of fast detection and high accuracy for the clinical sample, the manuscript some key issues should be addressed before its publication.
Reply: Thanks for your professional suggestions. We admit that the elaboration on the specificity of the sensor in the original manuscript was insufficient. We have reorganized the structure of the article to make the logic of each part clearer, particular attention has been paid to the "Materials and Methods" and "Conclusion and Discussion" sections. In the "Materials and Methods" section, we provided more detailed and precise information about the experimental procedures and principles. For the "Conclusion and Discussion" section, we have strengthened the analysis and interpretation of the results. Meanwhile, we supplemented experimental data and improved the article description in response to the reviewer's suggestions. All revisions are marked in blue.
Comments 1: The sensing principle of the sensor should be clarified.
Response 1: Thank you very much for your valuable suggestion. We do agree that the sensing principle of the sensor is not clearly presented in the original manuscript. In the revised version, we provided the theoretical background of the sensing principle. As in Lines 61-69, “Among them, graphene field effect transistor sensor (GFETs), a sensor using graphene as an active semiconductor material, is rapidly developing for biosensing applications. GFETs mainly rely on the principle that when biomolecules interact with the graphene surface, the electronic state of graphene is altered. This change leads to a shift in the Fermi level and, consequently, a change in the position of the Dirac point. By detecting this Dirac point shift, the detection and analysis of biomolecules can be achieved. [18-20]. The char-acteristic curves of GFETs have a good linear relationship, which reflects the real-time current signal response of the device to electrochemical and electrophysical reactions oc-curring on the graphene surface, and the response increases with the analyte concentra-tion[21, 22]”. Similarly, additions are made to the “Materials and Methods” section on Lines 135-140 regarding the principles of testing. All revisions are marked in blue. By providing a more detailed the sensing principle, we aim to make our work more understandable and reliable.
Comments 2:AFM testing can not confirm that the antibody probes are well immobilized on the surface of graphene. Please give the related characterizations, such as fluorescence labeling, EIS, and so on.
Response 2: Thank you for your insightful comments. We sincerely apologize for the absence of fluorescence labeling and EIS characterization data regarding the binding of antibody probes to the graphene interface in our work. However, as demonstrated in our previously published paper in Nat Biomed Eng. 2022 Mar;6(3):276 – 285(doi: 10.1038/s41551-021-00833–7), X-ray Photoelectron Spectroscopy (XPS) and Transfer Curve Characterization are also commonly used as powerful evidence for detecting probe modification on the graphene interface. For XPS, significant changes in the relative content of elements before and after modification can prove that the probe has been successfully modified onto the sample surface. In the case of transfer curve characterization, the introduction of the probe alters the surface charge distribution or energy-level structure of the system, thereby influencing the injection and transport of carriers. This results in a positive or negative shift in the threshold voltage of the transfer curve. In this study, we utilized XPS and transfer curve characterization to prove the successful modification of the antibody probe. It is evident from the transfer curve characterization results that the Dirac point (VDirac) of the NPAb-GFETs positively shifted 8 - 20 mV after the immersion of PASE and Influenza A virus nucleoprotein antibody (Figure 2a). This is indicating that the antibody probes are interacting with the graphene surface effectively. In addition, we have also utilized XPS (X-ray Photoelectron Spectroscopy) and analyzed the N 1s peak (as shown in Figure 2b). The appearance and changes in the N 1s peak can be attributed to the presence of the antibody probes, as the antibody contains nitrogen-containing functional groups. The observed N 1s peak and its characteristics are indicative of the successful immobilization of the antibody on the graphene surface, as the nitrogen atoms in the antibody contribute to this peak. In the Results section 3.1 in Lines 166-170, we revised it “The Dirac point (VDirac) of the NPAb-GFETs positively shifted 8~20 mV after the immersion of PASE and Influenza A virus nucleoprotein antibody (Figure 2a). This is attributed to the π-π interaction between PASE and graphene, which proves the correctness of the functionalization. After the immobilization of mAbs, the N 1s peak obtained by X-ray photoelectron spectroscopy shifted in both intensity and position (Figure 2b)”. Similarly, corresponding supplements have been made in the Materials and Methods section in Lines105-110 and 124-128. Thank you again for your attention and guidance on our work.
Comments 3: The author claimed that “The Dirac point (VDirac) of the NPAb-GFETs negatively shifted 8~20 mV after the immersion of PASE and Influenza A virus nucleoprotein antibody (Figure 2a)”. However, from Figure 2a, The Dirac point (VDirac) of the NPAb-GFETs shows a positive shift (toward high gate voltage direction). And the corresponding explanation is no correct. The reason of Dirac point shift is complicate. Please combine the sensing principle of the sensor, give the reasonable explanation.
Response 3: We sincerely apologize for the mistake. You are correct that the Dirac point (VDirac) of the NPAb-GFETs shows a positive shift (toward the high gate voltage direction) after the immersion, not a negative shift as originally stated. In the revised manuscript, we corrected the description accordingly, “The Dirac point (VDirac) of the NPAb-GFETs positively shifted 8~20 mV after the immersion of PASE and Influenza A virus nucleoprotein antibody (Figure 2a)” in Lines166. Regarding the Dirac point shift, we recognize that the reason for the shift is indeed complex. In light of your suggestion, we combined the sensing principle of the sensor to provide a more reasonable and comprehensive explanation. In Lines 61-69, “Among them, graphene field effect transistor sensor (GFETs), a sensor using graphene as an active semiconductor material, is rapidly developing for biosensing applications. GFETs mainly rely on the principle that when biomolecules interact with the graphene surface, the electronic state of graphene is altered. This change leads to a shift in the Fermi level and, consequently, a change in the position of the Dirac point. By detecting this Dirac point shift, the detection and analysis of biomolecules can be achieved. [18-20]. The char-acteristic curves of GFETs have a good linear relationship, which reflects the real-time current signal response of the device to electrochemical and electrophysical reactions oc-curring on the graphene surface, and the response increases with the analyte concentra-tion[21, 22]”. Similarly, corresponding supplements have been made in the Materials and Methods section in Lines in Lines105-110 and 124-128.
Comments 4: The Figure 2d is not cited in the manuscript.
Response 4: Thank you very much for bringing this to our attention. We apologize that we did not show Figure 2d explicitly in the original manuscript. We would like to clarify that the figure is indeed cited in the manuscript, where we state “The successful antibody functionalization can also be characterized by AFM (Figure 2c) and Raman results (Figure 2d and Figure S3)” in Lines 172.
Once again, thank you for your constructive comments, which will help us to further refine and improve our manuscript.

Round 2
Reviewer 3 Report
Comments and Suggestions for Authors
All question replies are ok.
Author Response

(The authors gave the same response as above.)
